# Impact of Anticoagulation and Sample Processing on the Quantification of Human Blood-Derived microRNA Signatures

**DOI:** 10.3390/cells9081915

**Published:** 2020-08-18

**Authors:** Marion Mussbacher, Teresa L. Krammer, Stefan Heber, Waltraud C. Schrottmaier, Stephan Zeibig, Hans-Peter Holthoff, David Pereyra, Patrick Starlinger, Matthias Hackl, Alice Assinger

**Affiliations:** 1Department of Vascular Biology and Thrombosis Research, Center of Physiology and Pharmacology, Medical University, Schwarzspanierstrasse 17, 1090 Vienna, Austria; marion.mussbacher@meduniwien.ac.at (M.M.); waltraud.schrottmaier@meduniwien.ac.at (W.C.S.); david.pereyra@meduniwien.ac.at (D.P.); 2TAmiRNA GmbH, Leberstrasse 20, 1110 Vienna, Austria; teresa.krammer@tamirna.com (T.L.K.); matthias.hackl@tamirna.com (M.H.); 3Department of Physiology, Center of Physiology and Pharmacology, Medical University, Schwarzspanierstrasse 17, 1090 Vienna, Austria; stefan.heber@meduniwien.ac.at; 4AdvanceCor GmbH, Fraunhoferstraße 9A, 82152 Planegg, Germany; zeibig@advancecor.com (S.Z.); holthoff@advancecor.com (H.-P.H.); 5Department of Surgery, Medical University of Vienna, General Hospital, Spitalgasse 23, 1090 Vienna, Austria; patrick.starlinger@meduniwien.ac.at

**Keywords:** plasma, microRNAs, anticoagulation, platelets, biomarkers

## Abstract

Blood-derived microRNA signatures have emerged as powerful biomarkers for predicting and diagnosing cardiovascular disease, cancer, and metabolic disorders. Platelets and platelet-derived microvesicles are a major source of microRNAs. We have previously shown that the inappropriate anticoagulation and storage of blood samples causes substantial platelet activation that is associated with the release of platelet-stored molecules into the plasma. However, it is currently unclear if circulating microRNA levels are affected by artificial platelet activation due to suboptimal plasma preparation. To address this issue, we used a standardized RT-qPCR test for 12 microRNAs (thrombomiR^®^, TAmiRNA GmbH, Vienna, Austria) that have been associated with cardiovascular and thrombotic diseases and were detected in platelets and/other hematopoietic cells. Blood was prevented from coagulating with citrate–theophylline–adenosine–dipyridamole (CTAD), sodium citrate, or ethylenediaminetetraacetic acid (EDTA) and stored for different time periods either at room temperature or at 4 °C prior to plasma preparation and the subsequent quantification of microRNAs. We found that five microRNAs (miR-191-5p, miR-320a, miR-21-5p, miR-23a-3p, and miR-451a) were significantly increased in the EDTA plasma. Moreover, we observed a time-dependent increase in plasma microRNAs that was most pronounced in the EDTA blood stored at room temperature for 24 h. Furthermore, significant correlations between microRNA levels and plasma concentrations of platelet-stored molecules pointed towards in vitro platelet activation. Therefore, we strongly recommend to (i) use CTAD as an anticoagulant, (ii) process blood samples as quickly as possible, and (iii) store blood samples at 4 °C whenever immediate plasma preparation is not feasible to generate reliable data on blood-derived microRNA signatures.

## 1. Introduction

Over the last decade, circulating microRNAs (miRNAs) have emerged as powerful biomarkers to predict and diagnose cardiovascular and thrombotic disease [1]. The number of studies emphasizing the clinical impact of these small, single-stranded non-coding RNAs is constantly rising, highlighting their enormous potential, by combining precision medicine and high-throughput quantitative analysis.

Platelets are the second most abundant cell type in the peripheral blood and play important roles in hemostasis, angiogenesis, wound healing, and immunity. Although platelets are anucleate cells, they contain disproportionately high amounts of miRNAs relative to their low level of protein synthesis [2]. In fact, platelets are currently estimated to contain about 750 different miRNAs [3]. These platelet-derived miRNAs can be detected in circulation where they significantly contribute to the pool of miRNAs in the plasma. The majority of plasma-derived miRNA is either bound to proteins or encapsulated in vesicles (e.g., extracellular vesicles, liposomes, and lipoproteins) and hence effectively protected from degradation by endogenous RNases [4]. Thus, circulating miRNAs are considered biochemically stable and show superior performance as biomarkers when compared to mRNAs [5]. We could recently show that accurate analysis of platelet-derived molecules in human plasma requires appropriate anticoagulation and standardized protocols to avoid the in vitro activation/degranulation of platelets [6]. Furthermore, a recent study by Mitchell and colleagues emphasized the importance of double centrifugation for the generation of platelet-poor plasma to avoid the contamination of the residual platelets [7]. In contrast to many platelet-derived molecules, which are released via the exocytosis of α-granules, miRNAs are released via microvesicle shedding, which represents a different and less studied mechanism. Furthermore, the very nature of miRNAs as nucleic acids may provide them with different stability and vulnerability to external influences such as anticoagulant and storage conditions than α-granule-derived molecules, which are, for the most part, proteins. To date, it is unclear if and to which extent pre-analytical variables such as the choice of anticoagulant, temperature, storage, and centrifugation time are essential for reducing the inter- and intra-study variability of plasma miRNA levels, and these have to be kept in mind when developing standardized protocols for miRNA detection in order to achieve high diagnostic accuracy while preserving clinical routine applicability.

To address this issue, we assessed in this study how miRNAs are affected by the choice of the (i) anticoagulant, (ii) storage temperature, and (iii) time until sample processing. We selected a panel of 12 miRNAs that have previously been associated with platelet activation and cardiovascular diseases and are expressed by different cell types (endothelial cells, platelets, and leukocytes). An overview of the selected miRNAs is given in Table 1. miR-197, miR-150, and miR-223 are among the most highly expressed miRNAs in platelets and platelet microparticles and are significantly downregulated upon dual anti-platelet therapy (aspirin/dipyridamole or clopidogrel) in patients with carotid atherosclerosis [8,9]. Additionally, plasma levels of miR-223 are decreased in patients with essential thrombocytopenia [10]. miR-23a is highly abundant in platelets [11] but correlates neither with the degree of platelet activation nor with clopidogrel resistance [12].

The plasma levels of miR-191 were found to positively correlate with C-reactive protein and pro-inflammatory cytokines in patients with diabetes mellitus type 2 [13]. Circulating miR-320 is significantly upregulated in patients with deep vein thrombosis [14]. miR-24 levels are unique for leukocytes, and the plasma levels of miR-24 are decreased in patients with abdominal aortic aneurysm [15]. Circulating miR-21 plasma levels are positively associated with recurrent venous thromboembolism [16] and increased in plasma samples contaminated by residual platelets [7].

miR-27a represents a positive regulator of inflammation [17] and is upregulated in macrophages upon exposure to LPS [18]. miR-126 is highly expressed by endothelial cells, endothelial cell apoptotic bodies, and platelets. It plays an important role in the regulation of vascular integrity, angiogenesis, and wound repair [19,20], and decreased miR-126 plasma levels are associated with the development of type 2 diabetes [21]. Moreover, miR-126 is reduced in patients receiving antiplatelet therapy [8]. miR-28 is increased in the plasma of patients with active chronic inflammatory bowel disease [22]. miR-451 is highly expressed by erythrocytes and acts as a negative regulator of fatty acid-induced inflammation [23].

In addition, multivariable regression models predicting miRNA levels based on plasma concentrations of the platelet-stored molecules platelet factor 4 (PF4), thrombospondin 1 (TSP-1), and P-selectin (CD62P) allowed us to quantify the impact of in vitro platelet activation on changes in measured miRNA levels.

## 2. Materials and Methods

### 2.1. Study Cohort of Healthy Volunteers

Six healthy volunteers (for demographic details, see Appendix A) were free of any medication for at least 2 weeks and gave their informed consent. Venous blood was collected from the antecubital vein using a 24G needle and indicated tubes containing anticoagulants. The study was performed in the morning hours (9:00–11:00 a.m.) in a non-fasted state, was approved by the Human Ethics Committee of the Medical University of Vienna (EK237/2004), and complied with the Declaration of Helsinki.

### 2.2. Plasma Preparation

Depending on the experimental set-up for plasma preparation, blood was drawn into pre-chilled (4 °C) or room temperature (RT) citrate–theophylline–adenosine–dipyridamole (CTAD), 3.8% sodium citrate, or dipotassium ethylenediaminetetraacetic acid (EDTA) tubes and stored at RT or 4 °C, respectively, until processing. Blood samples were centrifuged at the indicated time-points at 1000× *g* and 4 °C for 10 min. After this first centrifugation step, the plasma supernatant was transferred to a new vial and subjected to a second round of centrifugation at 10,000× *g* and 4 °C for 10 min to guarantee the removal of all cellular components. The supernatant was stored in aliquots at −80 °C until further use.

### 2.3. RNA Extraction and qPCR Analysis

The thrombomiR^®^ microRNA analysis kit (TAmiRNA, Vienna, Austria) was applied to isolate total RNA, including small RNAs, from 100 μL of plasma, which was diluted to 200 µL using nuclease-free water. To improve precipitation, glycogen was added to the aqueous phase prior to ethanol precipitation. RT-qPCR analysis was performed in single-replicate reactions according to the instructions provided by the kit. Reverse transcription was performed in a 10 µL reaction with 2 µL of total RNA input. The reverse transcription consisted of polyadenylation and was performed at 42 °C for 60 min followed by heat inactivation at 95 °C for 5 min. The resulting cDNA was diluted 1:20 for qPCR amplification in 10 µL reactions in primer-coated 96-well plates using green fluorescent dye (miGreen^®^, TAmiRNA, Vienna, Austria). Per sample, 13 microRNAs and 3 controls were analyzed in independent 10 µL reactions using locked-nucleic acid (LNA) enhanced target-specific miRNA primer pairs (which allow the discrimination of mature miRNAs from immature pre-miRNAs). The control assays were performed with synthetic spike-in controls added at the steps of RNA extraction (UniSp4), cDNA synthesis (cel-miR-39), and PCR amplification (UniSp3), which allow the determination of the yield, homogeneity, and overall robustness of the RNA extraction, complementary DNA synthesis, and qPCR amplification. PCRs were performed in a LightCycler 480 II (Roche, Mannheim, Germany) according to the following program: 95 °C, 2 min activation, followed by 45 cycles of 95 °C for 10 s and 60 °C for 60 s. Cq values were calculated using the 2nd derivative maximum method available in the Roche software. Melting curve acquisition was performed after 95 °C for 10 s, using continuous acquisition between 55 and 99 °C with a ramp of 0.11 °C/s and 5 acquisitions per °C.

Hemolysis was assessed using the ratios of miRNA-23a-3p and miRNA-451a [24], and positive samples were excluded from the analysis. RNA yield and assay variability were assessed on the basis of RNA, cDNA, and PCR spike-ins (Appendix A). The normalization of the microRNA Cq values was performed using the RNA spike-in control as an internal standard [25]. Delta Cq values (ΔCq) were calculated by subtracting the microRNA Cq values from the RNA spike-in Cq values according to the following formula: delta Cq value (ΔCq) = Cq(RNA-spike) − Cq(miRNA).

### 2.4. Determination of Plasma Concentrations

The plasma concentrations of sCD62P, TSP-1, and PF4 were analyzed using commercially available enzyme-linked immunosorbent assay (ELISA) kits (Quantikine; R&D Systems, Minneapolis, MN, USA) according to the manufacturer’s instructions.

### 2.5. Statistical Analysis

Data are presented as medians with interquartile ranges and were analyzed using one way and two-way ANOVA with Tukey or Sidak correction, respectively. The proportion of the variance of miRNA Cq values that could be explained by platelet activation markers was estimated by multiple linear regression analyses. To assess to which extent sample storage duration affected Cq values, (partial) linear correlation coefficients were calculated. A detailed description of the analyses can be found in the online Appendix A. All statistical analyses were performed using IBM SPSS Statistics 26; graphs were generated by Graphpad Prism 8.3.0. *p*-values < 0.05 were considered statistically significant.

## 3. Results

### 3.1. Anticoagulation Affects Blood-Derived miRNA Signatures

To assess whether the choice of anticoagulant during blood collection affected the plasma levels of miRNAs, we compared selected miRNAs that have been associated with alterations of platelet function (see Table 1 for details) between citrate–theophylline–adenosine–dipyridamole (CTAD), sodium citrate, and dipotassium ethylenediaminetetraacetic acid (EDTA) collection tubes (Figure 1). This representative panel of 12 miRNAs was selected from the literature based on (i) association with platelets and (ii) regulation in various cardiovascular/thrombotic diseases. The analytical assay variability was reduced by using a combination of synthetic spike-in controls (Appendix A).

We observed that the plasma levels of miR-191-5p, miR-320a, and miR-21-5p were significantly increased in EDTA plasma compared to those in CTAD plasma (Figure 1, middle panel). Moreover, the erythrocyte-specific miR-451a was significantly lower in both CTAD and citrate plasma than in EDTA plasma (Figure 1, lower panel). Overall, the observed inter-individual variation was significantly lower when CTAD was used as an anticoagulant (Appendix A). Detailed subject-specific visualization of data can be found in the Appendix A.

### 3.2. In Vitro Platelet Activation Increases Variability of microRNA Signatures

To unravel if the observed fluctuations of miRNA levels were associated with in vitro platelet activation, we estimated the proportion of miRNA level variance that could be explained by the plasma concentrations of PF4, TSP-1, and sCD62P (Figure 2). We found that PF4, which exclusively derives from platelets, could explain up to 15% of the variance in the miRNA abundance (Figure 2A), e.g., for miR-320a (Figure 2B) and miR-451a (Figure 2C). When we also accounted for TSP-1 and sCD62P levels (Figure 2D), up to 30% of the variance in abundance could be observed for the two miRNAs (Figure 2E,F). However, other miRNAs showed low or no correlation with platelet-derived plasma markers.

### 3.3. Storage Time Affects Blood-Derived miRNA Levels

Often, immediate plasma preparation might not be feasible under clinical conditions and blood samples are stored for several hours before further processing. To test whether the storage time has an effect on the levels of circulating miRNAs, we analyzed plasma after immediate preparation (30 min) and after 2, 6, and 24 h of storage at 4 °C (Figure 3). In CTAD, citrate, and EDTA samples, we observed an inverse correlation of miRNA levels over time, which was visible as a drastic drop in the correlation after 2 h (Figure 3A, Appendix A). Moreover, the effect of anticoagulation became more pronounced when we analyzed single miRNA ∆Cq levels over time (Figure 3B), resulting in a highly significant (*p* < 0.0001) difference between CTAD and EDTA plasma for the miRNAs miR-191-5p, miR-320a, miR-126-3p, and miR-21-5p. Moreover, milder but still-significant differences could be observed between the CTAD and citrate plasma for the miRNAs miR-191-5p and miR-126-3p and between the citrate and EDTA plasma for miR-191-5p, miR-320a, miR-21-5p, and miR-451a. The miRNAs miR-223-3p (*p* = 0.0006), miR-150-5p (*p* < 0.0001), miR-320a (*p* = 0.0002), miR-21-5p (*p* < 0.0001), miR-24-3p (*p* = 0.0034), and miR-451a (*p* = 0.0036) were significantly affected by storage time (Appendix A). However, an association between storage time and the anticoagulant could only be observed with miR-21-5p (*p* = 0.0235). Moreover, we observed that miR-223-3p, miR-150-5p, miR-320a, and miR-24-3p remained stable over storage for 6 h but significantly increased after 24 h. By contrast, miR-191-5p and miR-451a were already significantly elevated after 2 h of storage at 4 °C.

### 3.4. Storage of Blood at RT Causes A Significant Increase in miRNA Levels after 24 h

Next, we were interested in whether the observed impact of storage time on circulating miRNA levels was also observable upon incubation at RT. Therefore, we assessed miRNA levels in the plasma with different anticoagulants immediately after preparation (30 min) or after the storage of blood for 2, 6, and 24 h at RT (Figure 4 and Appendix A). Similar to what is shown in Figure 3, we observed an inverse correlation of the miRNA levels over time for all the anticoagulants tested (Figure 4A). Moreover, the storage of blood samples at RT revealed the same significant differences for miR-191-5p, miR-320a, miR-21-5p, and miR-451a as observed at 4 °C (Figure 4B). However, whereas the effect of incubation at RT was less pronounced for miR-126 3p, we observed that certain miRNAs were solely upregulated when stored at RT. This effect was most prominent after 24 h of incubation using EDTA as an anticoagulant and involved miR-223-3p, miR-197-3p, and miR-27-3p.

When we analyzed the effect of temperature (RT vs. 4 °C) on immediate plasma preparation (Appendix A), we found that temperature had a minor effect on circulating miRNA levels. However, miR-451a and miR-320a levels were affected by both temperature and anticoagulation: miR-451a levels decreased at RT (*p* = 0.0088 for temperature), an effect that was most prominent in the citrate plasma and associated with anticoagulation (*p* = 0.058 for interaction). By contrast, an increase in miR-320a levels was observed in the CTAD plasma when it was processed at RT.

### 3.5. Hemolysis Affects Blood-Derived miRNA Signatures

Another potential confounder for blood-derived miRNA signatures is the hemolysis that occurs due to the destruction of red blood cells upon inappropriate blood collection and/or plasma preparation (Figure 5). To account for this, we spiked CTAD plasma with different amounts of lysed red blood cells (RBCs, 0.016%, 0.125%, and 2%) and measured their impact on plasma miRNA levels. Whereas mild hemolysis (0.016% RBCs) had no effect on plasma miRNAs, we observed that the addition of 0.125% RBCs led to a minor but significant increase in the levels of miR-150-5p, miR-191-5p, miR-320a, miR-21-5p, miR-27-3p, and miR-451a. By contrast, the addition of 2% RBCs severely affected all analyzed plasma miRNA.

## 4. Discussion

With the discovery of miRNAs by Lee and colleagues in 1993, a new field of research has evolved, altering longstanding dogmas of gene regulation and bearing great diagnostic and therapeutic potential [36,37]. Circulating miRNA signatures have been identified as unique and powerful biomarkers not only for cancer but also for a variety of other diseases such as cardiovascular disease. However, to gain reliable results, which are essential for personalized medicine and to compare clinical studies, appropriate sample processing is mandatory. Indeed, many studies have focused on the preparation and normalization of blood-derived miRNAs [38]. Nevertheless, only a few studies have paid attention to plasma preparation itself [7,39]. In the present study, we addressed the issue of the preanalytical processing of blood for the analysis of circulating miRNAs and aimed to elucidate the role of anticoagulation, sample storage temperature, and time until plasma preparation. Based on a literature research, we choose a panel of 12 miRNAs that have been associated with platelets and thrombotic diseases and determined the effect of the anticoagulant, temperature, and storage time on the plasma levels of these miRNAs. We found that miRNA levels were significantly elevated in plasma samples that had been prepared from EDTA-anticoagulated blood and that this effect worsened over time and with storage at room temperature. Given the association between anticoagulation and storage time, the inhibition of platelet activation seems to be a crucial variable for the evaluation of robust miRNA signatures. This was further underlined by the correlation of miRNAs with the plasma concentrations of platelet-stored molecules.

Therefore, we strongly recommend to (i) use CTAD as an anticoagulant, (ii) process blood samples as quickly as possible, and (iii) store samples at 4 °C whenever immediate plasma preparation is not feasible. Moreover, we want to raise scientific awareness of the fact that optimal sample preparation is not only important in the cardiovascular and thrombotic fields; platelets store a multitude of miRNAs that are associated with angiogenesis and are therefore relevant for tumor growth and metastasis alike [19,29,30]. Although we focused on the cardiovascular role of the analyzed miRNAs, most of them were previously associated with different types of cancer [40,41]. The underlying reason for this could be found not only in the multiple targets of the miRNA but also in the versatile nature of platelets and their role in disease development and progression. This is emphasized by Best and colleagues, who could elegantly demonstrate that so-called “tumor-educated platelets” could be used to detect cancer and to monitor its progression [42]. As a consequence, our findings are relevant not only for the evaluation of plasma miRNAs but also for platelet isolation, as inappropriate anticoagulation would, vice versa, lead to alterations in the platelet-stored miRNA pool.

Our findings indicate that inappropriate anticoagulation is a confounder in the analysis of blood-borne miRNA signatures, which has a stronger impact than the stability of the miRNA itself, as we did not observe any decline in the abundance of miRNAs over the observed 24 h time period. This could be explained by the association of miRNAs with plasma proteins/lipoproteins or by the protective environment of platelet microvesicles, as miRNAs can be released either by granules or upon the formation of microvesicles. Both processes take place at the same time and are dependent on platelet activation. In this context, sodium citrate and acid-citrate-dextrose (ACD) have been recommended as the optimal anticoagulants for the study of microvesicles. Here, we have shown that citrate-containing anticoagulants (including CTAD) indeed cause the lowest degree of artificial in vitro platelet activation [6]. The beneficial effect of CTAD is probably achieved by the unique combination of citrate with theophylline, adenosine, and dipyridamole, which additionally prevent platelet activation by regulating the levels of cAMP and cGMP. Of note, the selected miRNA panel is not exclusively contained in platelets. Rather, most of the miRNAs investigated can be found in various hematopoietic cells, including highly abundant erythrocytes and various types of leukocytes, which could be activated by the type of anticoagulation alike. This argument is also supported by the finding that the spike-in of erythrocytes has a pronounced impact on plasma miRNA levels. Although we see a clear correlation between platelet-stored molecules and miRNA levels, this could also be secondary due to interplay between several cell types in the blood. The likelihood of cross-activation between different cell types is supported by the highly increased plasma levels of the erythrocyte-specific miRNA, miR-451a, in the EDTA plasma. Moreover, it is also conceivable that platelets take up circulating miRNAs and protect them from degradation in their open canalicular system or their granules. Such miRNAs, even if they are not expressed by platelets per se, would still be affected by artificial platelet activation due to inappropriate plasma preparation.

We want to point out that our miRNA panel is clearly limited by its low number and is able to cover neither all miRNAs abundant in blood cells nor all miRNAs relevant for cardiovascular disease. However, it is likely that the observed effects can be expanded to a broad range of miRNAs, even if they have not been addressed in this study. Therefore, this study is unable to provide a complete picture of circulatory miRNAs but emphasizes the importance of scientific awareness of the preclinical validation and consideration of appropriate anticoagulation. Moreover, we want to point out that the study was performed with the plasma of young, healthy, non-fasted blood donors and a limited sample size (*n* = 6). Further studies are warranted to allow the generalization of these results to patient cohorts. However, based on the results of our study, anti-platelet medication could represent a potential confounder for the determination of circulatory miRNA levels as has also been suggested by others [8].

Besides the importance of choosing an appropriate anticoagulant, we want to stress that it is likely that preparation of serum instead of plasma will have an even more profound effect on the content of circulatory miRNAs. In the course of serum preparation, both platelets and coagulation factors are maximally activated, resulting in the release of platelet-stored molecules and the induction of microvesicle shedding. Therefore, the serum levels of platelet-stored miRNAs do not reflect in vivo platelet activation/inhibition but rather mirror a mixture of miRNAs released by several different activated blood cells. Some contradictory results obtained by different research groups might be explained by the serum/plasma discrepancy; however, this is only valid for studies focusing on miRNAs that are associated or could be found in platelets or other blood cells. Special focus should be given to the parameters used for the validation of plasma/serum preparation; some studies follow the “the higher the better” concept, assuming that higher circulatory miRNA levels resemble more accurate detection. However, when it comes to the plasma/serum preparation of platelet-related miRNAs, the opposite is the case, as artificially high plasma levels can mask biologically relevant differences between patient cohorts. This conclusion is supported by other studies showing miRNAs to be differentially expressed in plasma and serum [43].

## 5. Conclusions

In summary, our data show that plasma preparation (the anticoagulation strategy, storage time, and temperature) impacts the abundance of miRNAs, most likely as a consequence of the in vitro activation of platelets and other blood cells. As miRNAs are steadily gaining importance as diagnostic tools and might soon be key determinants in personalized medicine, preanalytical shortcomings have to be minimized in order to focus on biologically relevant differences. We aim at raising awareness of this critical issue and highly recommend the use of CTAD as an anticoagulant for the preparation of plasma suitable for the analysis of blood cell-derived miRNA signatures.

## Figures and Tables

**Figure 1 cells-09-01915-f001:**
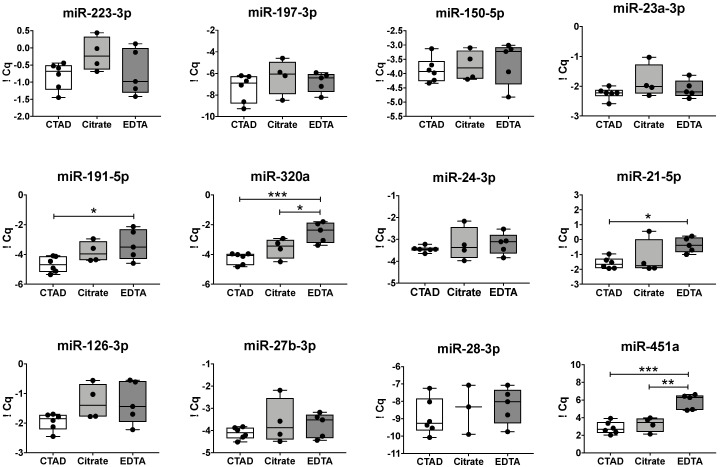
Plasma levels of miRNAs vary between different anticoagulants. Plasma miRNA levels were determined in 6 healthy individuals and are expressed as spike-in normalized quantitative cycles (∆Cq). Plasma was prepared from citrate–theophylline–adenosine–dipyridamole (CTAD) (white), citrate (grey), or ethylenediaminetetraacetic acid (EDTA) (dark grey) blood at 4 °C within 30 min of drawing blood. Significant differences were analyzed using one-way ANOVA with Tukey correction and are depicted as * *p* < 0.05, ** *p* < 0.01, and *** *p* < 0.001.

**Figure 2 cells-09-01915-f002:**
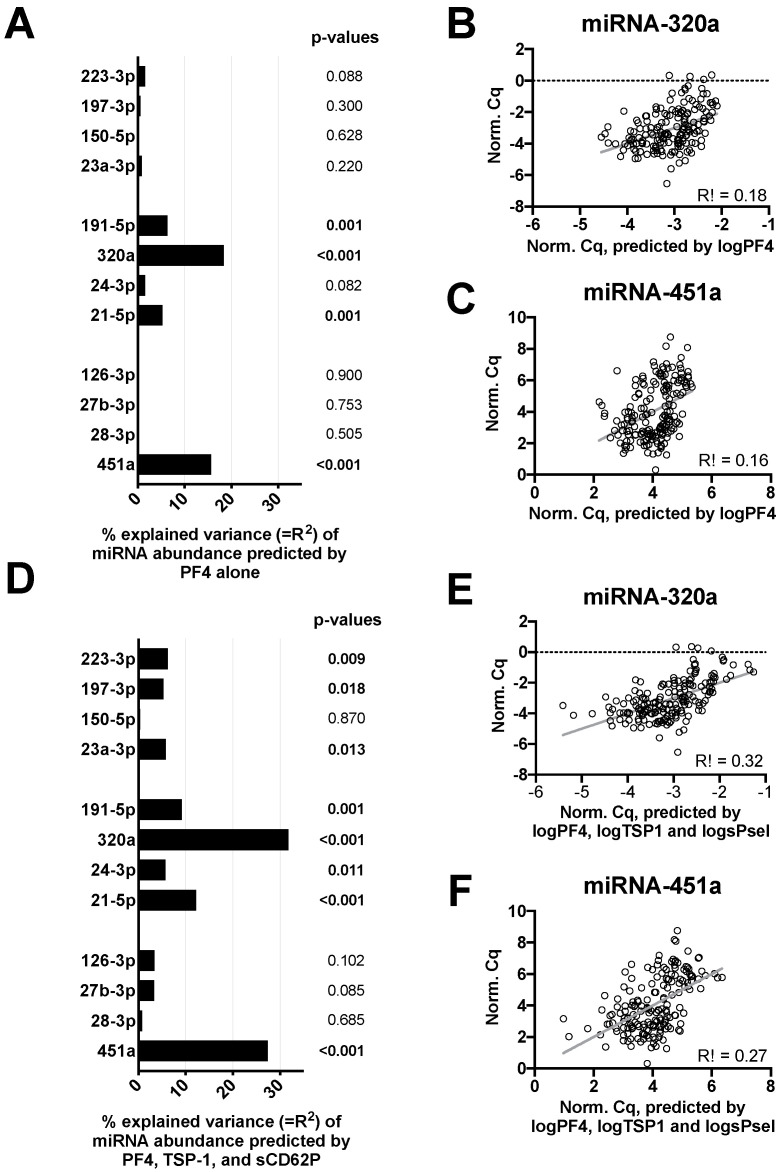
Dependence of miRNA abundance on platelet activation markers. To assess whether and to which extent the measured miRNA abundance might be affected by in vitro platelet activation, the variance of the miRNA values that could be explained by platelet activation, i.e., the R^2^ value, was determined. A theoretical value of 0% of explained variance (R^2^ = 0) would indicate that the respective miRNA is completely independent of platelet activation, whereas a value of 100% would indicate that the respective miRNA could entirely be explained by platelet activation in terms of a perfect correlation. Thus, miRNAs with higher percentages are more likely to be affected by in vitro platelet activation when quantified in plasma samples. (**A**) Variance percentage (=R^2^) of each miRNA that can be explained by PF4 alone. *p*-values refer to the hypothesis that PF4 alone explains miRNA variance. (**B**) Predicted vs. measured miR-320a ∆Cq values with corresponding R^2^ and r values. (**C**) Corresponding scatterplot of miR-451a. (**D**) Variance percentage (=R^2^) of each miRNA that can be explained by PF4, TSP-1, and sCD62P simultaneously. *p*-values refer to the hypothesis that PF4, TSP-1, and sCD62P simultaneously explain the miRNA variance. (**E**,**F**) Scatter plots showing predicted vs. measured values for miR-320a and miR-451a quantitative cycle (∆Cq).

**Figure 3 cells-09-01915-f003:**
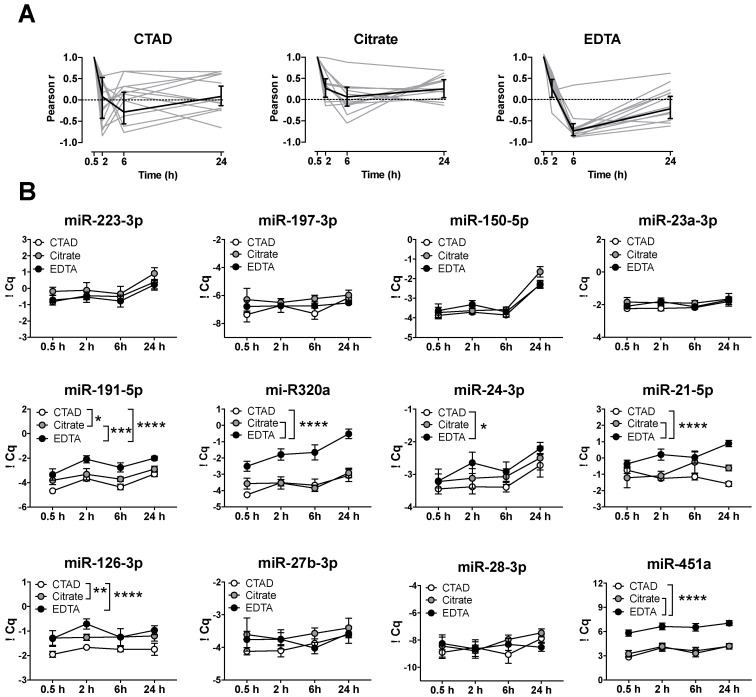
Time-dependent differences in circulating miRNA levels. (**A**) Pearson correlation coefficients between timepoint 0.5 h and all other timepoints for all miRNAs after storage at 4 °C. Each thin grey line represents one of 12 miRNAs; the bold lines represent means; error bars are 95% confidence intervals. Pearson correlation coefficient (r) of +1 indicates that the respective time point gives the same result as the 0.5 h time point; values of 0 would indicate that the values of that time point were unrelated to the values obtained at 0.5 h, and negative values indicate inverse values. (**B**) Blood from 6 healthy donors was prevented from coagulating with CTAD (white), citrate (gray), or EDTA (black) and stored at 4 °C for 0.5 (immediate plasma preparation), 2, 6, or 24 h until plasma preparation. Data are represented as means with standard errors. Significant differences were determined by two-way ANOVA using Sidak’s multiple comparison test and are depicted as * *p* < 0.05, ** *p* < 0.01, *** *p* < 0.001, and **** *p* < 0.0001.

**Figure 4 cells-09-01915-f004:**
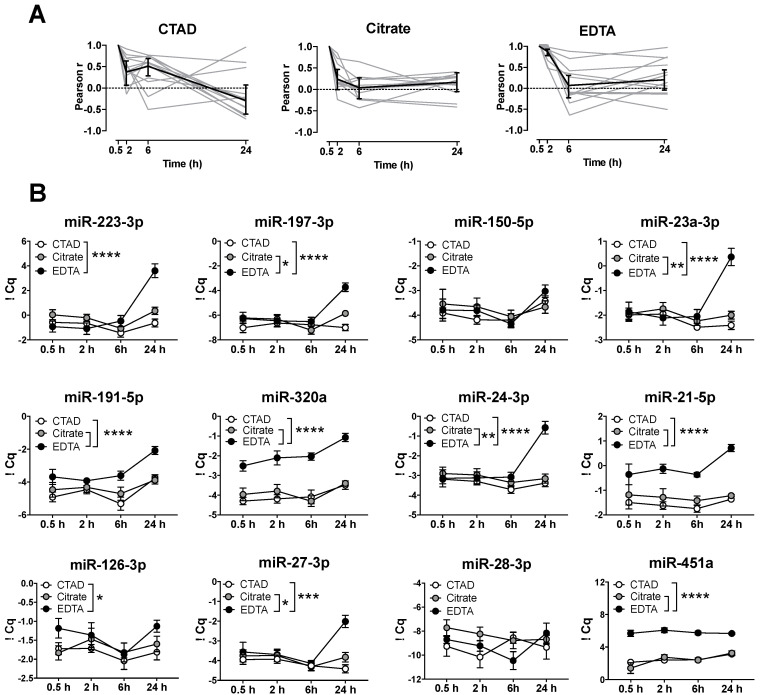
Time-dependent differences in circulating miRNA levels at room temperature (RT). (**A**) Pearson correlation coefficients between timepoint 0.5 h and all other timepoints for all miRNAs after storage at RT. (**B**) Blood from 6 healthy donors was anticoagulated with CTAD (white), citrate (gray), or EDTA (black) and stored at room temperature (RT) for 0.5 (immediate plasma preparation), 2, 6, or 24 h until plasma preparation. Data are represented as means with standard errors. Significant differences were determined by two-way ANOVA using Sidak’s multiple comparison test and are depicted as * *p* < 0.05, ** *p* < 0.01, *** *p* < 0.001, and **** *p* < 0.0001.

**Figure 5 cells-09-01915-f005:**
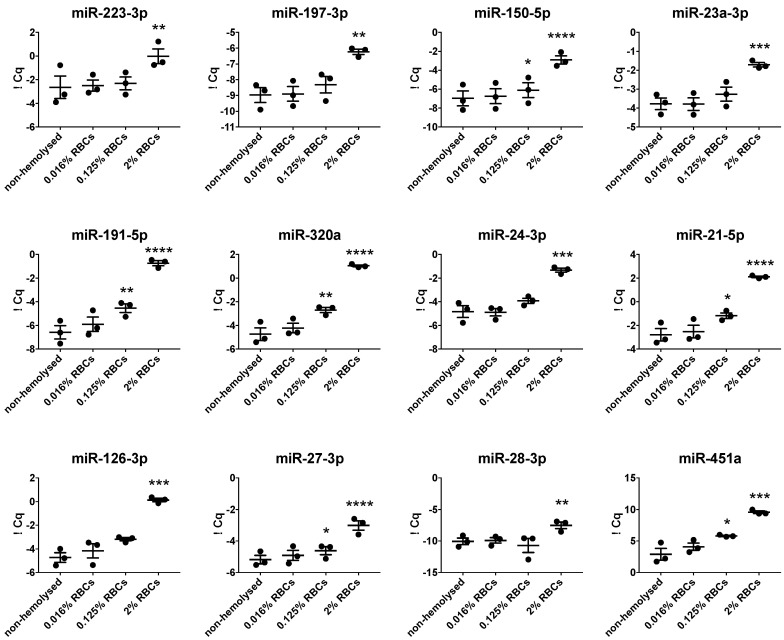
Impact of hemolysis on plasma miRNA levels. Plasma was spiked with red blood cells (RBCs, 0.016%, 0.125% and 2%) and miRNA levels were measured subsequently. Significant differences were determined by one-way ANOVA using Dunnett’s multiple comparison test and were depicted as * *p* < 0.05, ** *p* < 0.01, *** *p* < 0.001, and **** *p* < 0.0001.

**Table 1 cells-09-01915-t001:** Overview of selected miRNAs.

miRNA	Source	Function	Reference
miR-223-3p	platelets, hematopoietic cells	associated with decreased responsiveness to clopidogrel treatment in patients with coronary artery disease, miRNA-223 knockout in mice are still controversial	[26]
miR-197-3p	platelets, hematopoietic cells	predication of cardiovascular death in patients that suffered from symptomatic coronary artery disease	[27]
miR-150-5p	platelets, leukocytes	platelet maturation, decreased after switch from clopidogrel to ticagrelor	[9]
miR-23a-3p	platelets, hematopoietic cells	repressor of megakaryocyte development and differentiation	[28]
miR-191-5p	endothelial cells, platelets	identified as inhibitor of blood vessel development	[29]
miR-320a	cardiomyocytes, endothelial cells	cardiomyocyte apoptosis in cardiac ischemia/reperfusion injury, suppression of proliferation and migration of endothelial cells	[30]
miR-24-3p	macrophages, smooth muscle cells, platelets	inverse correlation with aortic aneurism	[15]
miR-21-5p	endothelial cells, smooth muscle cells, platelets, cardiomyocytes	increased in patients with acute myocardial infarct and angina pectoris	[31]
miR-126-3p	endothelial cells, platelets	vessel integrity, angiogenesis, wound repair, apoptosis	[19]
miR-27-3p	endothelial cells, adipocytes, platelets	adipogenesis, regulation of endothelial–mesenchymal transition, thrombin-induced synthesis of thrombospondin-1	[32]
miR-28-3p	cardiomyocytes, platelets	regulate the thrombopoietin receptor, marker for the diagnosis of pulmonary embolisms, promotes myocardial ischemia	[33]
miR-451a	erythrocytes	associated with hemolysis and erythropoiesis	[34,35]

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
