# Peer review of "Impact of Anticoagulation and Sample Processing on the Quantification of Human Blood-Derived microRNA Signatures"

_cells, 2020, doi:10.3390/cells9081915_

Round 1

Reviewer 1 Report

1. Were participants fasted before blood collection? Was blood collected at the same time of day for each participant? If not, please comment on the effect these variables may have had on your results, and whether this limits the generalisability of your findings. 

2. Suppl Table 1: please describe what the age values indicate (presumably mean and range) in the table.  

3. How did you determine that the inter-individual variation was lower for CDAT?

4. On page 6 you state that "To investigate whether the differences in miRNA levels observed due to different 180 anticoagulation, storage temperature and duration might be the result of associated in vitro platelet 181 activation, we estimated the proportion of miRNA..." However, at this point in the results you have only demonstrated an effect of anticoagulation, not storage temperature or duration.

5. How were spike-ins used for QC? Was there an upper Cq cut-off? (Fig S1)

6. How did you determine RNA purity?

7. Please complete a thorough spell-check of the MS for typographical and grammatical errors.

Author Response

We wish to thank the Reviewer for careful evaluation of our manuscript and the helpful suggestions to improve the quality of our study. Please find attached the point-to-point reply. 

Reviewer 2 Report

In the study by Assinger and co-workers, the authors assessed the impact of different pre-analytical plasma processing variables on levels of extracellular miRNAs. Specifically, the authors studied the impact of anticoagulation, sample storage temperature, and time until plasma preparation on RT-qPCR analysis of 12 miRNAs in blood from six young volunteers. These miRNAs were selected based on known associations with cardiovascular and thrombotic diseases. The authors found that miRNA levels were significantly elevated in plasma samples that had been prepared from EDTA-anticoagulated blood and that this effect worsened over time with storage at room temperature. Furthermore, they provide evidence that levels of extracellular miRNAs correlate with plasma concentrations of platelet-stored molecules, supporting the notion that the extent to which platelets become activated during plasma processing influence miRNA profiles.

This study provides strong support for previously published work showing the importance of variables associated with plasma processing in the assessment of extracellular miRNA profiles. There are now a large number of reports describing extracellular miRNA profiles as biomarkers for disease, but this work have been difficult to translate into the clinic because of variations in methodology and issues with reproducibility. The study by Assinger and co-workers identifies important causes of variability and poor reproducibility, and it provides guidance on how these issues can be overcome. However, their findings are expected and consistent with current recommendations for studies of extracellular miRNAs. It is not clear how much impact their findings will have on the field. In addition, further clarification of some of the experiments and conclusions should be provided.

Specific comments/questions:

  1. Current recommendations indicate that first few cc’s of blood should be wasted when blood is drawn for extracellular miRNA studies. Please clarify whether this was done in this study.

  1. The authors used a double centrifugation protocol to remove cellular components. Did they use any methods (such as flow cytometry) to verify that platelets were adequately removed with this protocol? Residual platelets in these samples could certainly impact miRNA levels.

  1. The authors used three different spike-ins. Please clarify which spike-in was used for normalization of miRNA values. Did the authors determine whether addition of glycogen had an effect on miRNA or spike-in levels?

  1. Please provide data for miR-23a to -451a ratios in samples used for this study.

  1. Figure 1 – inter-individual variation was reported to be smaller with CTAD. Wouldn’t it be expected that there is inter-individual variability, given different ages and gender of study participants? It is already accepted in the field that EDTA is not optimal for extracellular miRNA studies; others have observed increased generation of red blood cell derived microvesicles in EDTA anticoagulated blood. However, the authors should provide discussion as to why CTAD is better than citrate. Why do they think choice of anticoagulant only affects some miRNAs? Also, for this figure, it would be helpful if the authors could distinguish miRNA levels for each participant so the reader could see how miRNA levels for a given participant differ under the different anticoagulation conditions.

  1. Figure 2 – it is not clear why non platelet miRNAs are most affected by platelet activation. The authors suggest that non platelet miRNAs might have been taken up by platelets and released during activation. What is the evidence for this? Since PF4 exclusively derives from platelets, why does the variance in miRNA abundance change when TSP-1 and sCD62P levels are added to the analysis?

  1. The experiment for Figure 3 compares two variables – anticoagulant and time to plasma preparation. Panel B indicates statistical comparisons between different anticoagulants but statistical comparisons of different times for each anticoagulant are not clear, although they are described in the manuscript. Data for miR-223 is not shown in the figure. Is each point in panel B the mean miRNA value for the six participants? Please provide discussion for why there is an inverse correlation of miRNA levels at early time points that appear to change at later time points.

  1. Figure 4 - Questions regarding statistical comparisons and time point data for Figure 3 apply to Figure 4.

  1. Figure 5 – Please clarify how much RNA was present in each dilution of RBC lysate and how this amount of spike-in RNA compared to the total RNA in the plasma samples. Please clarify what lysed RBC percentage refers to – is this percent protein? Also, please clarify that this experiments involved a n = 3 for each spike-in. Does the finding that all miRNAs increase with the 2% spike-in mean that all of these cardiovascular miRNAs are in RBCs or is this an artifact associated with RBC proteins?

  1. A table or figure summarizing findings for each anticoagulant would be helpful.

  1. In the Introduction the authors describe several associations between miRNAs studied and clinical conditions (lines 79-84). Have the authors reviewed the methodology used in these studies? Do they think the plasma processing techniques in these studies was flawed? If so, should they be citing them as examples of miRNA biomarkers?

  1. In the Discussion, the authors indicate that the use of serum would be inappropriate for extracellular miRNA studies, but they did not study this and there are reports that miRNA levels in serum and plasma, when properly processed, are comparable. The authors should provide evidence to support their statement about serum-based studies or they should modify to account for why some in the field think serum is appropriate for extracellular miRNA studies.

Author Response

(The authors gave the same response as above.)

Reviewer 3 Report

This study by Mussbacher et al., examines a panel of platelet-enriched miRNAs in plasma from healthy donors under conditions of different anti-coagulants, and room temperature or cold storage for up to 24 hours, using qRT-PCR with an RNA spike-in control to determine relative miRNA levels. The goal is to determine whether these differences in sample preparation alter miRNA levels, an approach to blood-based biomarker studies which has been gaining interest in recent years. The authors show quantitative differences in some but not all of the miRNAs in the panel as a result of different anti-coagulant blood preparations and time in storage prior to sampling. They show data derived from whole blood indicating increased levels in platelet-derived proteins as a measure of storage-associated platelet activation, which the authors correlate with the anti-coagulant-dependent changes in miRNA levels. While the authors do not derive strong conclusions from the data regarding the sources of the altered miRNA levels, they instead use their data to provide specific recommendations for blood preparation for biomarker studies, specifically, CTAD as anti-coagulant, rapid processing, and refrigerated storage where longer times in storage are necessary.

The data overall are scientifically sound and reveal new information about how blood handling may affect miRNA biomarker detection. Thus, the study has merit that would be of interest to the readers of Cells. There are some minor concerns which should be addressed:

  1. The authors divide their panel of 12 miRNAs into three categories - platelet, cardiovascular, and inflammatory miRNAs - which appear to be somewhat arbitrary distinctions and have the unintended consequence of being misleading. The authors acknowledge that all the "thrombomiR" miRNAs they analyze are expressed in platelets, as well as other cells. It should be pointed out that some of the miRNAs excluded from the "platelets" category, such as miR-24, miR-191 and others, have been reported throughout the literature to be expressed at very high levels in platelets. Moreover, several of these miRNAs are also reported to have roles in platelet function, or conversely, "platelet miRNAs" have known roles in other functions such as miR-223-3p which is also associated with endothelial function that may derive from platelet-endothelial miRNA exchange. This review suggests revising the text to eliminate these arbitrary categories and combining the results for the full panel of platelet miRNAs. It is not possible to draw conclusions with respect to platelet/cardiovascular/inflammatory function with these data.
  2. A key detail is missing from the methods. It is unclear whether the qRT-PCR primers for miRNAs are specific for the cognate mature miRNAs, versus the immature pre-miRNAs which are also abundant in platelets and other cells. This should be clarified, and if the primers are not able to distinguish between these species, this should be acknowledged and discussed. Increased miRNAs have been observed in many studies in stored platelets, and these increases presumably derive from maturation over time by slicing of the hairpin pre-miRNAs by Dicer1 (except for miR-451, which the authors used in this study).
  3. The authors measure several platelet-derived components - PF4, TSP1 and soluble CD62P (P-selectin) as surrogate measures for increased platelet activation. It is also possible that the presence of these components in plasma represent release into plasma by platelets that have lysed due to handling or storage conditions, or platelets that have undergone apoptosis, a well-known progression particularly in stored platelets. In order to draw specific conclusions about platelet activation, the authors would need to provide direct evidence with isolated platelets from each condition, such as measurements of integrin activation with conformation-specific antibodies, surface exposure of CD62P and/or phosphatidylserine, etc.
  4. A nagging question that remains inadequately addressed is, if increased levels of detectable miRNAs under certain conditions are the result of platelet activation (or lysis, for that matter), why do some of the "platelet"-enriched miRNAs increase whereas others do not, relative to their overall abundance in platelets? Recent literature has indicated that release of miRNAs in microvesicles by activated platelets is roughly similar regardless of the platelet agonist. The authors were careful not to over-interpret their data with respect to platelets as the source of the increased miRNAs. Indeed, their data seem to more closely reflect miRNAs derived from erythrocytes. Still, this particular but fundamental point was not addressed specifically and it should be discussed, and conclusions regarding platelet miRNAs should be softened accordingly, absent additional data.
  5. Were storage times longer than 24 hours done and analyzed? Some of the miRNAs which did not show significant differences may in fact show significant alterations over longer time frames.
  6. At the risk of being a grammar scold, I feel the need to point out that p values should always be expressed as "<" rather than "=", which appears in some cases in the Results text.

Author Response

We wish to thank the Reviewer for careful evaluation of our manuscript and the helpful suggestions to improve the quality of our study. We highly appreciate the in-depth-platelet-knowledge of the Reviewer, who raised important and highly relevant questions.

Please find attached the point-to-point reply.

Round 2

Reviewer 2 Report

In the revised version of the manuscript by Assinger and co-workers, the authors have addressed concerns/questions raised in the previous review of the manuscript. These include concerns/questions about blood collection, spike-in normalization, residual cells and hemolysis, assessment of mature versus pre-miRNA, assessment of inter-individual variation in miRNA levels, relative contribution of activation/lysis of platelets versus other cells in miRNA variability, and why only certain miRNAs appear to be affected by anticoagulant, plasma storage temperature, and plasma processing time. Their study indicates that these variables are important for assessing extracellular miRNA levels, particularly those for miRs-191-5p, -320a, -21-5p, and -451a, and observed differences in extracellular miRNA levels with different anticoagulants are related to activation/lysis of platelets and other blood cells.

The responses by the authors and the additional data have resolved reviewer concerns about some of the experiments and the authors’ conclusions. While it is clear that EDTA should not be used for plasma miRNA profiling studies, it will likely require a larger study to verify that CTAD is significantly better than citrate for these kind of studies.